# Special Techniques of Adjuvant Breast Carcinoma Radiotherapy

**DOI:** 10.3390/cancers15010298

**Published:** 2022-12-31

**Authors:** Iveta Kolářová, Bohuslav Melichar, Jaroslav Vaňásek, Igor Sirák, Jiří Petera, Kateřina Horáčková, Denisa Pohanková, Zuzana Šinkorová, Oldřich Hošek, Milan Vošmik

**Affiliations:** 1Department of Oncology and Radiotherapy, Faculty of Medicine in Hradec Králové and University Hospital Hradec Králové, Charles University, 500 05 Hradec Králové, Czech Republic; 2Faculty of Health Studies, Pardubice University, 532 10 Pardubice, Czech Republic; 3Department of Oncology, Faculty of Medicine and Dentistry, Palacky University and University Hospital Olomouc, 779 00 Olomouc, Czech Republic; 4Oncology Centre, Multiscan, 532 03 Pardubice, Czech Republic; 5Department of Radiobiology, Faculty of Military Health Sciences, University of Defence, 500 01 Hradec Králové, Czech Republic

**Keywords:** breast cancer, adjuvant radiotherapy, boost, accelerated partial breast irradiation

## Abstract

**Simple Summary:**

Radiotherapy plays an important role in the multidisciplinary management of breast cancer. Historically, techniques for radiation treatment of breast carcinomas have been based upon two-dimensional planning and dose application. This treatment has been associated with a relatively high risk of chronic toxicity. Modern radiotherapy techniques have made it possible to achieve a better target volume coverage with a lower critical organ radiation burden. New fractionation methods have led to shorter treatment times while improving the therapeutic ratio. All of this has permitted a reduction in both acute and chronic toxicity, and led to improvements in treatment effectiveness. Currently, partial breast irradiation is used effectively in indicated cases. This can be achieved through utilizing various irradiation techniques and a number of fractionation schemes (e.g., partial breast irradiation and accelerated partial breast irradiation). Problematic issues are related to the application of adjuvant radiotherapy in patients immediately after breast replacement surgery.

**Abstract:**

Modern radiotherapy techniques are designed to permit reduced irradiation of healthy tissue, resulting in a diminished risk of adverse effects and shortened recovery times. Several randomized studies have demonstrated the benefits of increased dosage to the tumor bed area in combination with whole breast irradiation (WBI). Conventional WBI treatment following breast-conserving procedures, which required 5–7 weeks of daily treatments, has been reduced to 3–4 weeks when using hyperfractionated regimens. The dosage administration improves local control, albeit with poorer cosmesis. The method of accelerated partial breast irradiation (APBI) shortens the treatment period whilst reducing the irradiated volume. APBI can be delivered using intraoperative radiation, brachytherapy, or external beam radiotherapy. Currently available data support the use of external beam partial breast irradiation in selected patients. Modern radiotherapy techniques make it possible to achieve favorable cosmesis in most patients undergoing immediate breast reconstruction surgery, and studies confirm that current methods of external beam radiation allow an acceptable coverage of target volumes both in the reconstructed breast and in the regional lymphatic nodes.

## 1. Introduction

Breast carcinoma is the most common malignancy in women. Treatment has been substantially improved with the introduction of modern surgical procedures and new options for systemic treatment, as well as through a deeper understanding of the biological characteristics of the disease. 

The systemic therapy for breast cancer is currently determined by tumor molecular characteristics. Based on the expression of hormone receptors and human epidermal growth factor receptor (HER)-2, major tumor categories can be defined to guide the selection of hormonal therapy (tumors expressing hormone receptors), drugs targeting HER-2 (HER-2 overexpressing tumors), or cytotoxic chemotherapy (triple negative tumors) as the principal modalities of systemic treatment. Additional targets that define subgroups of patients with tumors harboring mutations of BRCA-1 or BRCA-2 genes or phosphatidylinositol-4,5-bisphosphate 3-kinase catalytic subunit alpha have recently been introduced.

The significant benefit gained from radiation cannot be underestimated. Radiotherapy (RT) plays an important role in the comprehensive management of patients with breast cancer. Historically, the techniques for radiation treatment of breast carcinomas were based upon two-dimensional planning and dose application. Such treatment methods were associated with a relatively high risk of chronic toxicity, including soft tissue fibrosis and lymphoedema, as well as lung and cardiac toxicity. In recent years, RT has fundamentally changed and its application has moved from two-dimensional to three-dimensional (3D) treatment, with the planning using simulation based upon computed tomography. This has led to significantly more precise treatment regimes, in particular with regard to the quality of coverage of target volumes and the radiation dose to critical organs. New fractionation methods have led to shorter treatment times, whilst at the same time improving the therapeutic ratio. All of this has made it possible to reduce both acute and chronic toxicity, while at the same time improving treatment effectiveness [1,2].

An important consideration when deciding on a therapeutic procedure is understanding the role of special radiotherapy techniques. One of the possibilities is partial breast irradiation, which may be administered using various irradiation techniques and fractionation schedules. Another issue is related to understanding in which situations an increased dose to the tumor bed (boost therapy) should be indicated. A third issue concerns the application of adjuvant radiotherapy in patients immediately after breast reconstruction.

In view of the changes that have taken place in RT for patients with breast carcinomas, the aim of this narrative review is to provide current information on present-day RT techniques to radiation oncologists and the other team members who participate in deciding on treatment methods. An electronic literature search was conducted in the PubMed and Web of Science databases for English articles published up to May 2022. Search terms used were “breast cancer radiotherapy,” “breast cancer,” “accelerated partial breast irradiation,” “partial breast cancer irradiation,” “boost,” “hypofractionated radiation therapy,” “breast reconstruction and radiation therapy,” and “brachytherapy.” The review focuses on radiotherapy techniques and dose/fractionation issues, but it does not include the topics of combining radiotherapy with systemic treatment, molecular biological predictors of treatment, radiobiological topics, or a number of other points.

## 2. Significance of Biological Subtypes of Breast Cancer for Radiotherapy

Originally, breast cancer subtypes were defined based on immunohistochemistry [3]. Subsequently, four molecular subtypes of breast cancer have been defined by analyzing complementary DNA (cDNA) chips containing 8102 genes [4], and different systemic treatment strategies have been outlined based on these classifications [5]. 

Triple negative breast cancer (TNBC) accounts for approximately 20% of all breast cancer cases. The absence of the three receptors significantly reduces options for targeted approaches in patients with TNBC [6]. It was originally believed that the locoregional recurrence of HER2-positive subtype and TNBC are similar, but for patients with overexpression of HER2, the use of trastuzumab significantly reduces the locoregional recurrence rate. On the other hand, TNBC is sensitive to chemotherapy, and so chemotherapy represents the principal treatment modality in these patients [7]. 

It is obvious that the different breast cancer subtypes may also differ in response to radiation, but the data that would allow diversification of treatment based on tumor phenotypes are lagging behind those which we have for systemic treatment. On the other hand, differences in tumor biology associated with tumor phenotype present an opportunity to investigate novel radiation therapy strategies. A cohort study investigating the selection of local treatment for 768 patients with early TNBC, with a mean follow-up of 7.2 years, showed that postoperative radiotherapy can minimize the risk of local recurrence of TNBC, emphasizing the importance of radiotherapy in local control of TNBC [8].

Through a meta-analysis comparing different breast cancer subtypes, a relative resistance of TNBC to radiotherapy has been reported [9]. Advances in the understanding of molecular mechanisms could result in improved therapies, including radiation treatment. For example, an important role may be played by the Notch1 protein in BRCA-1 related tumorigenesis [10]. 

## 3. Boost to the Tumor Bed

Factors predictive of ipsilateral breast tumor recurrence (IBTR) include younger age, higher grade, positive margins, and associated ductal carcinoma in situ (DCIS) [11].

The benefit of a boost to the tumor bed was evaluated by the Lyon and EORTC 22,881 trials. The Lyon trial evaluated 1024 breast cancer patients younger than 70 years of age treated with breast conserving surgery and subsequent adjuvant radiation (50 Gy in 20 fractions), who were randomized between boost (electron beam, 10 Gy in 4 fraction) and no boost treatments. After follow-up at 3 years, those patients treated with boost therapy had significantly lower local recurrence rates (3.6% vs. 4.5%, *p* = 0.044). Cosmesis was similar in both arms of the study, but telangiectasia was more common in patients treated with the boost regime [12].

The EORTC 22881 trial enrolled 5569 patients with stage T1-2, N0-1 breast cancer, aged 70 years or younger, and treated with lumpectomy and partial radiation therapy (50 Gy). Patients with negative margins (95% of the total population) were randomized to boost (16 Gy to tumor bed) or no boost treatments. Patients treated with boost had significantly lower local recurrence rates at 5 or 10 years (4% vs. 7%, *p* < 0.0001; 6% vs. 10%, *p* < 0.0001, respectively). Administering boost therapy reduced the number of salvage mastectomies by 41%. Distant metastasis-free survival and overall survival (OS) rates were similar in both arms. Patients of all age subgroups derived benefit from the boost regimen compared to WBI alone. The relative risk reduction was similar in all age subgroups, but the absolute reduction of risk was greatest in younger patients. On the other hand, administration of boost treatment did not improve disease-free survival (DFS) or OS rates, and was associated with a higher incidence of fibrosis and telangiectasia [11,13,14,15].

To summarize, in cases of invasive tumors, boost treatment is indicated for women aged 50 years of age or younger with positive margins, and for women between 51 and 70 years of age with high-grade tumors or positive margins. Boost therapy may be omitted in women over 70 years of age with low or intermediate grade tumors and negative margins (2 mm or larger). In the case of patients with DCIS, boost treatment is recommended for women 50 years of age or younger with high-grade tumors or positive margins (smaller than 2 mm) [16]. According to the GEC-ESTRO Breast Cancer Working Group, boost treatment may be omitted in women older than 50 years of age with ≤3 cm unicentric, unifocal tumors with no nodal involvement, with negative margins ≥2 mm, with no lymphovascular invasions or no extensive intraductal components, and not triple negative tumors. Boost therapy is indicated for other breast carcinoma patients on a case-by-case basis [17]. 

The preferred manner of application, which may be carried out either as a supplemental dose after terminating WBI, or via simultaneously integrated tumor bed boost treatment as a part of a hypofractionated WBI regime, is a matter of some deliberation. Use of the simultaneous integrated boost technique shortens adjuvant radiation therapy times without compromising clinical outcomes [18,19].

## 4. Hypofractionation Regimens of Breast Irradiation

Radiobiological studies have shown a relatively low alpha/beta ratio for breast carcinoma [20] which is supposed to have similar efficacy and toxicity using hypofractionation and conventional fractionation regimens. A number of trials have investigated hypofractionation regimens with whole breast irradiation. A Canadian study compared two fractionation regimens in patients with pT1-2, N0, margin-negative disease: either 42.5 Gy in 16 fractions and 22 days, or standard whole breast irradiation of 50 Gy/25 fractions and 35 days. No differences were observed in either acute skin toxicity or frequency of other complications. Long-term follow-up demonstrated comparable efficacy in both treatment arms [21].

The efficacy of hypofractionation regimens was evaluated through a meta-analysis of four randomized trials in 7,095 patients [22]. No differences in the risk of ipsilateral local recurrence or in breast cosmesis were found, and the risk of acute and/or late toxicity was reduced (risk ratio 0.21, 95% confidence interval 0.07–0.64).

Ten-year follow-up of the START-A and START-B trials demonstrated no significant difference between conventional radiotherapy and hypofractionation regimens in patients with pT1-3a, N0-1, margin-negative disease. Based upon results of the START-B trial, the current standard fractionation regimen in Britain is 40 Gy in 15 fractions [23].

The American Society for Radiation Oncology (ASTRO) guidelines recommend hypofractionated WBI for breast cancer patients without regard to age, tumor stage, or whether or not they have undergone chemotherapy. The appropriate scheme is hypofractionated WBI of 4000 cGy doses in 15 fractions or 4250 cGy in 16 fractions. The recommendation discusses factors that may or should affect the decision-making regarding fractionation, and states subgroups that are recommended or suggested for the use or omission of boost treatment, together with recommended dosages [16].

Hypofractionated regimens comprise the preferred method of radiotherapy in breast carcinoma treatment today [24]. The consensus of the European Society for Radiotherapy and Oncology (ESTRO) is that moderately hypofractionated radiotherapy can be offered to any patient for the irradiation of whole breast, chest wall (with or without reconstruction), and regional nodes. Ultrafractionation (26 Gy in 5 fractions) can also be offered for breast or chest wall radiotherapy (without reconstruction), either as a standard treatment or as part of a randomized study or prospective cohort [25]. Risk of late toxicity remains a subject for discussion and further long-term observation [16].

Further research is ongoing to test larger doses per fraction (>2.0 Gy). The results of two phase III trials (UK IMPORT HIGH and RTOG 1005) are not yet released, but may show whether dose escalation through 3D conformal RT or intensity-modulated concomitant boost after breast-conserving surgery can reduce RT side effects while sustaining or even improving treatment results in those patients having a greater risk of local recurrence [26,27].

## 5. Partial Breast Irradiation (PBI)

The rationale for PBI is based upon data demonstrating that most recurrences after breast-conserving surgery are observed in the tumor bed or its immediate vicinity, and, therefore, radiation limited to this region instead of the whole breast may eliminate the residual disease while achieving acceptable cosmesis and toxicity [28,29,30]. Moreover, a prolonged course of conventional WBI presents an obstacle to the wider use of breast conservation therapy [31]. A shorter treatment course (5 to 15 days), possible lower tumor repopulation, and potential for better cosmesis (depending upon the technique) are listed among the advantages of accelerated partial breast irradiation (APBI) [30]. PBI and APBI can be delivered using brachytherapy (Figure 1), intraoperative radiation, or external beam techniques [32,33,34,35,36].

### 5.1. Interstitial Brachytherapy

Interstitial brachytherapy is the APBI technique with the longest use experience [37,38]. A multi-catheter technique is used for interstitial breast brachytherapy. Flexible after-loading catheters are placed through the breast tissue surrounding the lumpectomy using a free-hand- or template- guided approach. The catheters are inserted at 1 to 1.5 cm intervals in several planes encompassing the surgical cavity, with consideration of a safety margin. A CT scan is used for the 3D treatment planning. After delineation of the estimated tumor bed (ETB), which includes the surgical scar, clips, and estimated tumor location, the clinical target volume (CTV) is defined, ensuring a safety margin of ≥20 mm in all directions. The safety margin is the sum of the free resection margin and the added radiation safety margin (at least 10 mm). The CTV is cropped to the chest wall (pectoral muscle or ribs) and 5 mm from the skin surface. The prescribed dose to the CTV is usually 34 Gy in 10 fractions, delivered twice daily (minimal target dose, MTD), where 90% of the CTV should be covered by ≥90% of the MTD, the maximum dose to the skin should be ≤MTD, the volume of 150% of the isodose (V150) should be kept below 50 cm^3^, and the dose nonuniformity ratio (DNR) should be ≤30% [39].

The earliest APBI trials were performed using interstitial brachytherapy at a time when the selection criteria for the patients were not so strict and the planning of dose distribution was less advanced. A large number of ipsilateral recurrences (37%) has been reported in the pilot trials of low dose rate brachytherapy [40].

Studies using newer brachytherapy techniques and better patient selection have reported a substantially lower recurrence incidence. A large randomized phase III trial comparing APBI using interstitial brachytherapy utilizing multiple catheters with whole breast irradiation included 1184 patients after breast-conserving surgery. A total of 551 patients were treated with conventional external radiation with a tumor-bed boost, while 633 patients were randomized to APBI with brachytherapy. After 5 years the number of recurrences in the APBI and external radiotherapy arms were 9 and 5, respectively. The cumulative incidence of local recurrence was 1.44% in the APBI arm and 0.92% in the external radiotherapy arm. The rate of grade 2 or 3 late cutaneous effects at 5 years was 3.2% with APBI compared to 5.7% with external radiotherapy (*p* = 0.08). The trial demonstrated the noninferiority of APBI with brachytherapy using multiple catheters compared to whole breast irradiation, as indicated by 5-year local control, DFS, and OS [41].

### 5.2. Intraoperative Radiotherapy 

#### 5.2.1. Intracavitary Brachytherapy

In May 2002, MammoSite^®^ (Cytyc Corporation, Marlborough, MA, USA) became the first FDA-approved intracavitary device [42]. A silicon balloon is connected to a catheter with a filling channel and a second channel for the radiation source. In the case of single channel applicator, the target volume for balloon-brachytherapy is defined as 1 cm around the balloon surface. The disadvantage of this is decreased conformity of irradiation in comparison with a multi-catheter technique, and a risk of higher skin doses, that can result in late skin toxicity. The solution is to use a multichannel applicator such as the Contura device (SenoRx, Inc, Aliso Viejo, CA, USA) or the SAVI device (Cianna Medical, Aliso, Viejo, CA, USA). In this case the CTV is defined in the same manner as for interstitial brachytherapy [43]. 

The device may be inserted during surgery to localize the postoperative cavity. The most robust data for APBI brachytherapy was sourced from the American Society of Breast Surgeons MammoSite^®^ breast brachytherapy registry trial, which reports a 5-year local recurrence rate of 3.8% and low toxicity [44,45]. 

#### 5.2.2. Electrons

Intraoperative radiotherapy using electrons allows for substitution of the conventional postoperative irradiation of the whole breast with an equivalent dose applied via a single radiotherapy session during the operation [32].

Patients within the intraoperative radiotherapy group were given a single dose of 21 Gy to the tumor bed during surgery. Those treated with external beam radiation were given a dose of 50 Gy in 25 fractions. 

This was an equivalence trial, and the equivalence limit defined in advance was local recurrence of 7.5% within the group undergoing intraoperative radiotherapy. The principal endpoint was the occurrence of ipsilateral breast tumor recurrence (IBTR). Overall survival was a secondary result. The principal analysis was by intention to treat. The 5-year event rate for IBTR was 4.4% (95% CI = 2.7–6.1) in the intraoperative radiotherapy group and 0.4% (0.0–1.0) in the external radiotherapy group (hazard ratio [HR] = 9.3; 95% CI = 3.3–26.3). Although the frequency of the IBTR rate in the patients treated with intraoperative radiotherapy was within the prior defined equivalence interval, it was significantly higher than in the external radiotherapy group, while the OS rate did not differ between the two groups.

### 5.3. Postoperative Partial Breast Irradiation 

This is a non-invasive technique with advantages over brachytherapy and intraoperative radiotherapy that include ease of accessibility and potentially a better dose homogeneity. In contrast with WBI, PBI is based upon irradiating a limited volume of tissue (Figure 2). External postoperative radiotherapy has many advantages in comparison with intraoperative procedures. The treatment is initiated with knowledge of the definite and complete histology, including the immunohistochemistry, and, in comparison with brachytherapy, it is less dependent upon the individual technical quality of the treatment’s execution. 

There are studies on PBI that support its acceptability in patients with low-risk breast cancer, either through using moderately hypofractionated radiotherapy (PBI) or through a lower number of fractions.

IMPORT LOW was a multicentric, randomized, controlled phase III noninferiority study that was carried out at 30 radiotherapy centers in the United Kingdom among women 50 years of age and older who underwent breast-conserving surgery for grade 1–3 unifocal invasive ductal adenocarcinoma, with tumor size 3 cm or less (pT1-2), zero to three positive axillar nodes (pN0-1), and minimal microscopic margins of 2 mm and greater. The patients were randomly divided (1:1:1) into groups with an applied dose of 40 Gy via the WBI technique (control group); 36 Gy via the whole breast technique and 40 Gy to part of the breast (group with reduced dose); or 40 Gy to part of the breast (group with reduced volume); always in 15 daily fractions. The median period of observation was 72.2 months (interquartile range: 61.7–83.2), and 5-year estimates for cumulative incidence of local relapse were 1.1% (95% CI = 0.5–2.3) for patients in the control group, 0.2% (0.02–1.2) in the reduced dose group, and 0.5% (0.2–1.4) in the PBI group. The estimated 5-year absolute differences in local relapse in comparison with the control group were −0.73% (−0.99 to 0.22) in the reduced dose patients and −0.38% (−0.84 to 0.90) in the PBI group. Noninferiority was confirmed both for irradiation by reduced dose and for PBI, and an equivalent or lower occurrence of side effects in the healthy tissue areas was observed [24].

Different fractionation regimens for external APBI have been tested, mainly 30 Gy in 5 fractions in 10 days, and 34 Gy in 10 fractions which was later changed to 38.5 Gy in 10 fractions [46]. This regimen was used at various centers and subsequently in a large randomized RTOG 0413/NSABP B-39 trial that evaluated the efficacy and toxicity of APBI [47]. 

In the RTOG 0413/NSABP B-39 study, the whole breast irradiation was administered in 25 daily fractions to a total dose of 50 Gy over 5 weeks, either with or without using a supplementary boost to the tumor bed; various techniques of APBI including brachytherapy and 3D conformal external radiotherapy were allowed. APBI was applied either as 34 Gy brachytherapy or as 38.5 Gy via an external beam in 10 fractions over 5 days (2 fractions per day), with the total duration of treatment not exceeding 8 days [48].

Analysis of the IBTR principal endpoint showed 90 events (4.6%) in the ABPI arm and 71 events (3.9%) in the WBI arm. APBI did not meet the criteria for equivalence to WBI in controlling IBTR following breast-conserving surgery.

Distant, disease-free, and overall survival rates did not differ between the groups. Analysis of the IBTR showed that recurrences in the tumor bed appeared in the APBI and WBI arms with equal frequency. In the APBI group, however, recurrences occurred more frequently in other parts of the breast (1.5% vs. 2.7%, HR = 1.99, CI-95% = 1.23–3.23). Exploratory analysis of the APBI methods suggested that the methods of brachytherapy applied were associated with a higher rate of recurrence (7.7% for multicatheter brachytherapy and 7.8% for single-entry brachytherapy) in comparison with external beam RT. The number of complications and the rate of second malignancies did not significantly differ.

The similarly designed RAPID (Randomized Trial of Accelerated Partial Breast Irradiation) study reached different conclusions. The RAPID study randomized patients treated with APBI using external irradiation with dose of 38.5 Gy in 10 fractions over 5 to 8 days (2 fractions daily) versus conventional WBI with a dose of 42.5 Gy in 16 fractions over 21 days or 50 Gy in 25 daily fractions.

Analysis of the principal endpoint, which was IBTR, showed 37 events (3.0%) in the APBI arm and 28 events (2.8%) in the WBI arm, which led to a risk ratio of 1.27 (CI 90% = 0.84–1.91). That result was below the prior defined 2.02 upper limit of the noninferiority margin. DFS, event-free survival, and OS rates also showed no differences between the treatment methods. Acute toxicity (grade ≥ 2) was greater in the WBI arm, affecting 47% of patients vs. 28% of patients in the APBI arm. Late toxicity (grade ≥ 2), however, occurred more frequently in the ABPI arm, affecting 32% of patients vs. 13% of patients in the WBI arm. In the APBI arm, the assessment of cosmesic effect by nurses in the 7th year after the treatment demonstrated an inferior outcome with the results stated as fair or poor in 36% of cases for APBI vs. 19% of cases for WBI. In the assessment by the patients themselves, cosmesis was fair or poor in 31% of APBI cases vs. 15% of WBI cases. In preventing IBTR, external beam APBI was noninferior to WBI [49].

A 2010 meta-analysis covering three randomized and 19 prospective nonrandomized trials with a minimum follow-up time of 4 years demonstrated that patients treated with APBI had a greater risk of local and axillary recurrence. Overall survival rates and the risk of distant metastases were similar [50]. The results of this comparison between APBI and WBI remain controversial, however, as the meta-analysis struggled with several problems, including differing selection criteria, differing target volume definition, and differing irradiation techniques. 

The principal problem related to APBI lies in the precise determination of the tumor region and consequent difficulty in localizing the target volumes, and particularly so in the case of brachytherapy [51,52,53,54].

Among the most prominent disadvantages of APBI are a lack of long-term data for some of the specific techniques used, the potential for unknown late effects when utilizing high doses per fraction, and the need for two treatments per day. Considering the excellent long-term results of the hypofractionation WBI alternative and the burden that twice-daily irradiation imposes upon both patients and staff (including physicians and physicists), APBI may not be appropriate in all situations [55,56].

Patients 50 years of age or older with small tumors and negative lymph nodes who are candidates for postoperative radiotherapy limited to the breast, and who fulfill the consensus criteria guidelines may be adequately treated using APBI. Patients with multicentric tumors, disease involving lymph nodes, tumor sizes greater than 3 cm, lymphovascular invasions, and pregnancy-associated breast cancers are not considered appropriate candidates for this approach. Other parameters that may not favor APBI include extensive DCIS, invasive lobular carcinoma negative hormone receptors, HER-2 positivity, and BRCA1/2 mutation, in particular when these are present in patients younger than 50 years of age [57].

We may conclude that PBI seems to be an acceptable alternative to WBI if applied to suitable risk groups (pT1, pN0, R2, HR+, nonlobular histology, age >50 years, without extensive DCIS). Techniques such as interstitial brachytherapy, radiotherapy with a modulated intensity of the beam via a dose of 30 Gy in 6 fractions over 2 weeks, or 3D conformal radiotherapy via a dose of 40 Gy (15 × 2.67 Gy) over 3 weeks may be applied [1].

## 6. Breast Reconstruction and Radiotherapy

The issue of using radiotherapy in patients who have undergone immediate breast reconstruction and in whom post-mastectomy radiation therapy (PMRT) is indicated remains controversial. Available data indicate that a majority of patients undergoing immediate reconstruction have an implant placed during the surgery [58]. Figure 3 presents an isodose plan for a patient with an inserted silicone implant.

Such patients may be treated with PMRT either while a temporary expander (PMRT-TE) is in place or after its exchange for a permanent implant (PMRT-PI). The timing for the insertion of the temporary implant and its replacement with a permanent implant is dependent upon chemotherapy administration. Irradiation with a temporary expander in place is associated with a greater risk of complications, but this can be explained by the short interval between radiotherapy and the exchange of the temporary expander for the permanent implant. By delaying the expander–implant exchange for at least 6 months after completing the postmastectomy radiation therapy, the risk of expander–implant failure can be significantly reduced [59].

Although both PMRT-PI and PMRT-TE are associated with certain risk of capsular contracture, favorable cosmesis is achieved in most cases. The risks associated with PMRT are nevertheless considered unacceptable by some authors, who instead recommend reconstruction using autologous tissue [60].

In contrast to some earlier data, more recent studies have confirmed that current methods of radiotherapy allow satisfactory coverage of target volumes both in the reconstructed breast and in the regional lymph nodes [61].

Although they do not surpass recommended limits, it has been demonstrated that greater doses to the lungs and heart can result from the irradiation of internal mammary lymph nodes after immediate reconstruction [62]. In patients receiving PMRT, bilateral implants do not compromise the target volume coverage nor increase the lung and heart dosages. In patients with implant-based reconstruction, be that unilateral or bilateral, the most crucial predictor of high lung and heart doses is treatment of the internal mammary nodes [63].

Among the most prominent approaches to resolving these problems is the modification of the contouring of target volumes, in order to better spare the critical organs. Current PMRT guidelines for patients with implants from the ESTRO Advisory Committee in Radiation Oncology Practice recommend, in selected cases, excluding the implant from the target volumes, as well as excluding parts of the dorsal chest wall from the CTV. The target volume then includes only the part of the tissue anterior to the implant and the pectoralis major muscle. Nevertheless, data from prospective trials that would support this practice remain lacking [64,65].

## 7. Cardiac Toxicity 

Cardiac toxicity illustrates the importance of the evaluation of long-term treatment results along with chronic toxicities. For example, Darby et al. reported in a population-based surveillance study of 2168 women post-RT that major coronary events had a linear dose response relationship with the mean cardiac dose. For every 1 Gy increase in mean heart dose, there was a linear 7.4% increase in the incidence of major coronary events.. Mean RT dose to the heart to ≥10 Gy resulted in a 116% increase in major coronary events [66].

In clinical practice, approaches that allow reduction of the radiation dose to the heart and lungs are often used. These techniques differ but include radiotherapy timing synchronized with the breathing cycle, prone patient positioning, or using modern radiotherapy technologies (i.e., IMRT or proton beam therapy), as well as reduction of the traditional target volumes (whole breast) to a more confined volumes (accelerated partial breast irradiation or intraoperative radiotherapy).

Breath holding regimes represent one of the most well studied cardiac-sparing techniques and can be offered to the majority of breast cancer patients, although patients with poor respiratory function or inability to sustain breath holding may not be treated using this technique. Breath holding techniques can also be combined with other approaches, including IMRT [67]. 

## 8. Effects of Breast Cancer Therapy on Cosmesis

A satisfactory cosmetic result is an important determinant of patient quality of life and contentment. The assessment of cosmetic results remains a complex issue that has so far not been settled in a satisfactory manner. The evaluation by the patient herself is subjective, and prone to reflect the patient’s personality traits but a subjective evaluation performed by a panel of experts is an expensive and time-consuming procedure.

Consequently, much effort has been dedicated to the creation of an objective assessment method, e.g., computer-based approaches such as Breast Cancer Conservative Treatment Cosmetic Results software (BCCT.core). This software is a free, semi-automated, and easy-to-use tool that provides a highly reliable and reproducible assessment of the esthetic outcome. The IMRT-MC2 trial compared three assessment methods. In this trial, patients were treated with breast-conserving surgery (BCS) and adjuvant radiotherapy. In total, 502 patients were randomized to a control arm (3-D-CRT-seqB) and an experimental arm (IMRT-SIB). The results indicated that an assessment by the BCCT.core software alone may not be sufficient to evaluate the esthetic outcome as perceived by the patients. The BCCT.core software is a valuable and reliable tool to measure objective asymmetries, but should be complemented by both physician assessment and patient self-assessment in order to include the subjective perception of esthetics.

Currently, satisfactory cosmesis is achieved in the majority of patients, but 30–40% of patients report being disappointed with the final cosmetic result following BCS. A fundamental determinant of favorable cosmesis is the surgical procedure itself. Breast asymmetry after BCS is significantly correlated with poor psychosocial functioning and fear of recurrence, which have a negative impact on QoL [68].

In addition to the surgical procedure, the cosmesis can be also affected by other therapeutic interventions. In an observational cohort study long-term therapeutic results were evaluated in 104 breast cancer patients who underwent BCS at least five years prior to the study. Cosmetic results were negatively influenced by sentinel node procedures, axillary lymph node dissections, chemotherapy, and hormonal therapy. Expert panel and objective BCCT.core software assessments resulted in similar outcomes. Three out of four patients have acceptable long-term results regarding their QoL and esthetic results [69].

## 9. Perspectives

In its current state of development, radiotherapy for the treatment of breast carcinoma aims to reduce the risk of the side effects and increase patient comfort. One approach is to omit radiotherapy where possible by identifying patients who might only minimally benefit from it. The ongoing EUROPA study compares adjuvant radiation therapy by the PBI technique with adjuvant endocrine therapy as per the local policy [70].

The completion of axillary lymph node dissection is not mandatory in patients with one or two sentinel lymph node metastases who will subsequently undergo WBI and systemic therapy, as the ACOSOG Z0011 and IBCSG 23-01 trials demonstrated no difference in recurrence rates after axillary lymph node dissection following sentinel node biopsy over a sentinel node biopsy alone in patients with both macroscopic and microscopic metastases [71,72].

The AMAROS trial that compared ALND versus radiotherapy for the axilla in patients with positive sentinel lymph node biopsy and negative clinical findings in regional lymph nodes did not report any difference in recurrence rates. The rate of lymphedema was doubled in the ALND arm (28% vs. 14%) [73]. 

The results of these investigations indicate the possibility of substituting ALND with radiotherapy in patients with positive sentinel lymph node and negative clinical findings.

New approaches are being sought to identify very low risk carcinomas in situ and invasive carcinomas. These include assessment methods using the biological, clinical, and histological markers of low risk in combination with individual assessment and selection of patients who could safely forego whole breast radiation therapy following breast-conserving surgery (BCS) [2,74].

Genomic testing is a promising way to identify patients not requiring adjuvant radiotherapy. This issue is the subject of the ongoing DEBRA study, which evaluates the significance of a particular test (Oncotype DX) in assessing suitability of postoperative irradiation. Low-risk patients in stage I and HR+/HER2− with Oncotype DX scoring of ≤18 are treated by adjuvant hormonal treatment and then randomized into study arms comparing outcomes with or without irradiation treatment [75].

The TAILOR RT study, meanwhile, focuses on assessing the significance of adjuvant irradiation of regional lymphatic areas in patients with low risk according to Oncotype DX testing. Patients with a recurrence score of ≤25 following BCS or mastectomy, with clear margins of excision, limited nodal involvement, ER+/HER2− status, and who have been treated with endocrine therapy are selected. In one arm of the study BCS patients are treated using WBI or no RT following mastectomy. In the other arm, WBI plus RT is administered to the regional nodes (supraclavicular, non-dissected axillary, and internal mammary) following BCS, or to the chest wall and regional nodes (supraclavicular, non-dissected axillary, and internal mammary) following mastectomy [76].

The possibility of using other tests is being investigated. The DCISionRT test was developed to predict the benefits of radiotherapy and for assessing the risks of relapse in women diagnosed with DCIS. Various studies have been conducted on this topic. Bremer et al. assessed the relationship between the decision score and the 10-year risk of invasive breast carcinoma or any ipsilateral breast episode, including invasive breast carcinoma or DCIS. The benefit of RT was assessed according to risk group and as a function of the decision score [77].

An external validation study assessed patients treated using BCS with or without radiotherapy. This study proved the capacity to identify patients having DCIS with a low risk of relapse and, therefore, likely to obtain minimal benefit from adjuvant RT [78].

Extremely short regimens of PBI are being tested in patients with early-stage breast carcinoma following breast-conserving surgeries. Due to progress in the radiotherapy technique, which substantially increased the conformity of irradiation, ultrashort fractioning schemes are being used during which the treatment is administered in one to five fractions [79].

Such treatment has proven very useful during the COVID-19 pandemic. A 2020 COVID-19 pandemic recommendation states that “delivery of radiotherapy in five fractions only for all patients requiring radiation therapy with node negative tumors that do not require a boost is recommended.” In the United Kingdom during the COVID19 pandemic, the use of 26 Gy in 5 fractions increased from 0.2% of all courses in April 2019 to 60.6% in April 2020 [80,81].

In patients after radical mastectomy who are undergoing breast reconstruction, the possibility of using moderately hypofractionated radiotherapy is also being studied with the aim of reducing toxicity and at the same time shortening the treatment time and its cost [82].

As with many other diagnoses, the possibility of performing stereotactic body radiation therapy (SBRT) in patients with oligometastatic disease is increasingly being utilized. This treatment is associated with a minimum of adverse complications. Patients with breast carcinoma benefit from this treatment due to favorable results in local control and possibly experience increased overall survival rates. Further randomized controlled studies are needed in order to determine the optimal combination of SBRT with system therapy [83,84].

The basic conditions for successful stereotactic radiotherapy are the treatment quality assurance, including correct indication and choice of dose and fractionation, correct treatment planning and dose distribution calculation, as well as technical equipment and documentation at the highest level [85].

## 10. Conclusions

Radiation therapy (RT) is one of the essential components in the management of breast cancer. It has a significant role to play in the comprehensive treatment of early-stage breast cancer. Advances in breast RT technology and treatment delivery techniques have fundamentally changed the options for optimizing the delivery of care. Significantly lower local recurrence rates are seen in selected patients. Hypofractionated RT may be suitable for whole breast, chest wall (with or without reconstruction), and nodal volumes in all patients. Recent research findings confirm that current methods of RT allow satisfactory coverage of target volumes, both in the reconstructed breast and in regional lymph nodes, without exceeding dosage limits in vital organs.

With the aim of reducing treatment toxicity, methods of partial irradiation have been introduced that achieve equivalent rates of local control in selected patients. A patient 50 years of age and older with a small tumor and negative lymph nodes, who is a candidate for postoperative radiotherapy confined to the breast, and who satisfies the consensus guidelines criteria may be satisfactorily treated using APBI. In the future, a further increase in the probabilities of achieving therapeutic effects, while at the same time reducing the risk of side effects from treatment, may be expected due to the possibilities of utilizing genomic testing.

## Figures and Tables

**Figure 1 cancers-15-00298-f001:**
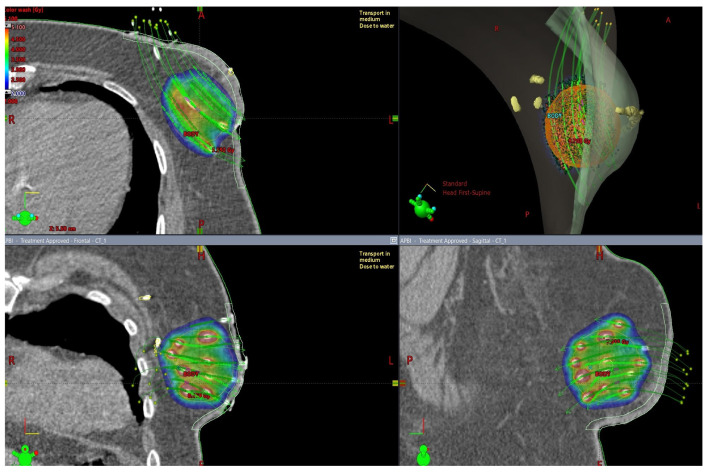
3D visualization of multicathether interstitial APBI technique from different viewing angles, delineation and dose calculation: interstitial catheters—white, CTV—orange, skin—light green, surgical clips and skin incision—yellow. Dose 3.4 Gy per fraction. Dose distribution presented by the color distribution: V150 (5.1 Gy)—red, V100 (3.4 Gy)—green, 2 Gy—blue.

**Figure 2 cancers-15-00298-f002:**
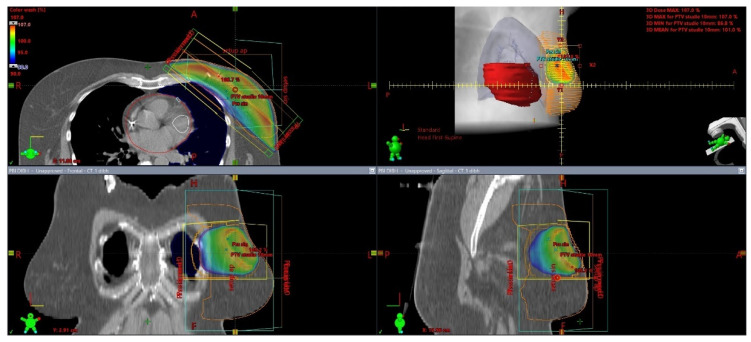
3D visualization of external beam PBI technique from different viewing angles, delineation and dose distribution: heart—red, left ventricle and left anterior descending artery—white, left breast—orange, left lung—blue. Photon dose distribution produced by LINAC is presented by the color distribution: 95—107% of prescribed dose; 40 Gy/15 fractions.

**Figure 3 cancers-15-00298-f003:**
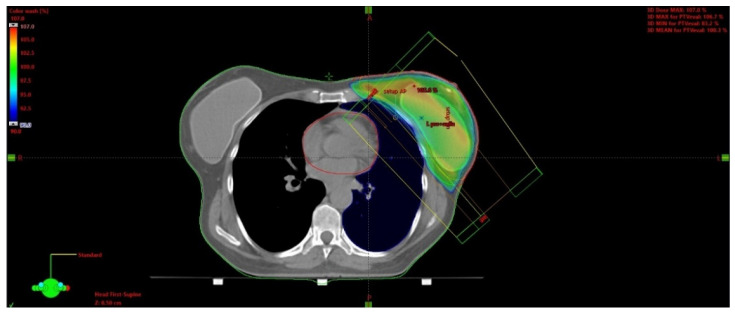
Breast irradiation after radical mastectomy with silicone implant insertion. A dose greater than 90% of the prescribed dose is displayed in color. The maximum dose in this slice is 105.6 % of the prescribed dose.

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
