# Peer review of "Special Techniques of Adjuvant Breast Carcinoma Radiotherapy"

_cancers, 2022, doi:10.3390/cancers15010298_

Round 1

Reviewer 1 Report

A very good review of breast cancer radiotherapy techniques, primarily for breast conservation, providing both historical experience and current recommendations. I have the following comments which could strengthen the manuscript.

Page 2, line 76: factors predictive of IBTR. I would add positive margins to this list.

Page 3, line 99. Positive margins in the younger women would be an indication for a boost as well.

Page 3, line 104. Indications for omission of a boost are important; I would reference this statement.

Figure 1 and Figure 2. A detailed description of these Figures should be presented in the Figure legend. To the non-radiation therapist, these figures are difficult to understand, in part because there are also no labels. The red on black print is basically unreadable. What is the animated figure supposed to represent?

Page 4, line 164. Please provide some technical information for the administration of interstitial irradiation. Are these all delivered with through-and-through catheter placement?

Page 5, line 181. For intraoperative irradiation, a brief description of delivery by balloon catheter vs. column irradiation would be helpful.

Cosmesis is one of the most important endpoints of breast conserving irradiation. I would include at least a paragraph discussing cosmetic results, both short term and long-term, and any future efforts to further improve outcome.

Page 9, Line 356. The concept of adjuvant irradiation of regional lymphatic areas is introduced. The irradiation of axillary nodal tissue in place of axillary dissection is an important current topic. I would include comments about this and highlight the Amaros trial as well.

A brief discussion of the methods to reduce cardiac exposure would be helpful.

The limitations to the review should be discussed.

Reviewer 2 Report

##Manuscript Title  "Special Techniques of Adjuvant Breast Carcinoma Radiotherapy"

1.The authors need to provide some basic introduction about types of breast cancer and which kind of breast cancer subtypes provide best results after radiotherapy.

2.Introduce importance breast cancer somatic and germline mutation and their role in breast tumorigenesis, high throughput sequencing and driver genes mutations in breast cancer and cite these articles (https://doi.org/10.1038/s41467-020-16936-9)

3. A clear table with types of breast cancer and radiation therapy, success rate, overall survival and disease-free survival rate are provide better information.

4. Is there any combination therapy provide betterment to the patients? The authors need to provide combination of radiation therapy, what are all the other treatment methods such as chemo and immunotherpy provide better results? discuss.

5.Is there any other biological factors correlated with Radio therapy benefit patients? 

6. Its better to provide TNBC and radio therapy and outcomes.

7. Also provide chemo+radio therapy with some case studies on breast cancer will provide better picture about the importance of this review.

8. The authors may consider to cut-short the long boring paragraph into informative table form.

Round 2

Reviewer 2 Report

Manuscript looks good and most of the suggested comments are addressed.